# Clinical Performance of the Consensus Immunoscore in Colon Cancer in the Asian Population from the Multicenter International SITC Study

**DOI:** 10.3390/cancers14184346

**Published:** 2022-09-06

**Authors:** Bernhard Mlecnik, Toshihiko Torigoe, Gabriela Bindea, Boryana Popivanova, Mingli Xu, Tomonobu Fujita, Shoichi Hazama, Nobuaki Suzuki, Hiroaki Nagano, Kiyotaka Okuno, Yoshihiko Hirohashi, Tomohisa Furuhata, Ichiro Takemasa, Prabhudas Patel, Hemangini Vora, Birva Shah, Jayendrakumar B. Patel, Kruti N. Rajvik, Shashank J. Pandya, Shilin N. Shukla, Yili Wang, Guanjun Zhang, Takayuki Yoshino, Hiroya Taniguchi, Carlo Bifulco, Alessandro Lugli, Jiun-Kae Jack Lee, Inti Zlobec, Tilman T. Rau, Martin D. Berger, Iris D. Nagtegaal, Elisa Vink-Börger, Arndt Hartmann, Carol I. Geppert, Julie Kolwelter, Susanne Merkel, Robert Grützmann, Marc Van den Eynde, Anne Jouret-Mourin, Alex Kartheuser, Daniel Léonard, Christophe Remue, Julia Wang, Prashant Bavi, Michael H. A. Roehrl, Pamela S. Ohashi, Linh T. Nguyen, SeongJun Han, Heather L. MacGregor, Sara Hafezi-Bakhtiari, Bradly G. Wouters, Giuseppe V. Masucci, Emilia Andersson, Eva Zavadova, Michal Vocka, Jan Spacek, Lubos Petruzelka, Bohuslav Konopasek, Pavel Dundr, Helena Skalova, Kristyna Nemejcova, Gerardo Botti, Fabiana Tatangelo, Paolo Delrio, Gennaro Ciliberto, Michele Maio, Luigi Laghi, Fabio Grizzi, Florence Marliot, Tessa Fredriksen, Bénédicte Buttard, Lucie Lafontaine, Pauline Maby, Amine Majdi, Assia Hijazi, Carine El Sissy, Amos Kirilovsky, Anne Berger, Christine Lagorce, Christopher Paustian, Carmen Ballesteros-Merino, Jeroen Dijkstra, Carlijn Van de Water, Shannon van Lent-van Vliet, Nikki Knijn, Ana-Maria Mușină, Dragos-Viorel Scripcariu, Francesco M. Marincola, Paolo A. Ascierto, Bernard A. Fox, Franck Pagès, Yutaka Kawakami, Jérôme Galon

**Affiliations:** 1INSERM, Laboratory of Integrative Cancer Immunology, 75006 Paris, France; 2Centre de Recherche des Cordeliers, Sorbonne Université, Université de Paris, 75006 Paris, France; 3Equipe Labellisée Ligue Contre le Cancer, 75006 Paris, France; 4Inovarion, 75005 Paris, France; 5Department of Pathology, Sapporo Medical University, Sapporo 060-8556, Japan; 6Division of Cellular Signaling, Institute for Advanced Medical Research, School of Medicine, Keio University, Tokyo 160-8582, Japan; 7Department of Translational Research and Developmental Therapeutics against Cancer, Yamaguchi University School of Medicine, Yamaguchi 755-8505, Japan; 8Department of Gastroenterological, Breast and Endocrine Surgery, Yamaguchi University Graduate School of Medicine, Yamaguchi 753-8511, Japan; 9Department of Surgery, Kindai University, School of Medicine, Osakasayama 589-0014, Japan; 10Department of Surgery, Surgical Oncology, and Science, Sapporo Medical University, Sapporo 060-8556, Japan; 11The Gujarat Cancer & Research Institute, Asarwa, Ahmedabad 380016, India; 12Institute for Cancer Research, School of Basic Medical Science, Xi’an 710061, China; 13Health Science Center of Xi’an Jiaotong University, Xi’an 710061, China; 14Department of Gastroenterology and Gastrointestinal Oncology, National Cancer Center Hospital East, Kashiwanoha, Kashiwa-shi 277-8577, Japan; 15Department of Pathology, Providence Portland Medical Center, Portland, OR 97213, USA; 16Institute of Pathology, University of Bern, 3008 Bern, Switzerland; 17Department of Biostatistics, M.D. Anderson Cancer Center, University of Texas, Houston, TX 77030, USA; 18Department of Medical Oncology, University Hospital of Bern, 3010 Bern, Switzerland; 19Pathology Department, Radboud University, 6500 HC Nijmegen, The Netherlands; 20Department of Pathology, University Erlangen-Nürnberg, 91054 Erlangen, Germany; 21Department of Surgery, University Erlangen-Nürnberg, 91054 Erlangen, Germany; 22Institut Roi Albert II, Department of Medical Oncology, Cliniques Universitaires St-Luc, 1200 Brussels, Belgium; 23Institut de Recherche Clinique et Experimentale (Pole MIRO), Université Catholique de Louvain, 1200 Brussels, Belgium; 24Department of Pathology, Cliniques Universitaires St-Luc, 1200 Brussels, Belgium; 25Institut de Recherche Clinique et Experimentale (Pole GAEN), Université Catholique de Louvain, 1200 Brussels, Belgium; 26Institut Roi Albert II, Department of Digestive Surgery, Cliniques Universitaires St-Luc Université Catholique de Louvain, 1200 Brussels, Belgium; 27Curandis, New York, NY 10583, USA; 28Department of Pathology, Laboratory Medicine Program, University Health Network, 11-E444, Toronto, ON M5G 2C4, Canada; 29Department of Laboratory Medicine and Pathobiology, University of Toronto, Toronto, ON M5S 1A8, Canada; 30Department of Pathology, Memorial Sloan Kettering Cancer Center, New York, NY 10065, USA; 31Princess Margaret Cancer Centre, Toronto, ON M5G 2C1, Canada; 32Department of Oncology-Pathology, Karolinska Institutet, Karolinska University, 17177 Stockholm, Sweden; 33Department of Oncology, First Faculty of Medicine, General University Hospital in Prague, Charles University, 12808 Prague, Czech Republic; 34Institute of Pathology, First Faculty of Medicine, General University Hospital in Prague, Charles University, 12808 Prague, Czech Republic; 35Department of Pathology, Istituto Nazionale Tumori IRCCS Fondazione G. Pascale, 80131 Naples, Italy; 36Colorectal Surgery Department, Instituto Nazionale Tumori IRCCS Fondazione G. Pascale, 80131 Naples, Italy; 37IRCCS Istituto Nazionale Tumori “Regina Elena”, 00128 Rome, Italy; 38Center for Immuno-Oncology, University Hospital, 53100 Siena, Italy; 39Laboratory of Molecular Gastroenterology, IRCCS Humanitas Research Hospital, Rozzano, 20090 Milan, Italy; 40Department of Medicine and Surgery, University of Parma, 43125 Parma, Italy; 41Department of Immunology and Inflammation, IRCCS Humanitas Research Hospital, Rozzano, 20090 Milan, Italy; 42Department of Biomedical Sciences, Humanitas University, Pieve Emanuele, 20072 Milan, Italy; 43Immunomonitoring Platform, Laboratory of Immunology, AP-HP, Assistance Publique-Hopitaux de Paris, Georges Pompidou European Hospital, 75015 Paris, France; 44Digestive Surgery Department, AP-HP, Assistance Publique-Hopitaux de Paris, Georges Pompidou European Hospital, 75015 Paris, France; 45Department of Pathology, AP-HP, Assistance Publique-Hopitaux de Paris, Georges Pompidou European Hospital, 75015 Paris, France; 46Department of Molecular Microbiology and Immunology, Oregon Health and Science University, Portland, OR 97239, USA; 47Department of Surgical Oncology, Regional Institute of Oncology, University of Medicine and Pharmacy “Grigore T. Popa”, 700115 Iaşi, Romania; 48Kite Pharma, Santa Monica, CA 90404, USA; 49Melanoma, Cancer Immunotherapy and Innovative Therapies Unit, Istituto Nazionale Tumori IRCCS Fondazione “G. Pascale”, 80131 Naples, Italy; 50Laboratory of Molecular and Tumor Immunology, Earle A. Chiles Research Institute, Robert W. Franz Cancer Center, Providence Portland Medical Center, Portland, OR 97213, USA

**Keywords:** Immunoscore, colon cancer, tumor microenvironment, immune response, classification, prognostic markers, risk stratification, T cell, MSI, Asian

## Abstract

**Simple Summary:**

Research interest in Immuno-oncology and the role of the adaptative immune system in the progression and prognosis of colon cancer (CC) is growing. In this study, we evaluated the prognostic value of the consensus Immunoscore in 423 patients with AJCC/UICC-TNM stages I–III CC from Asian care centers. Immunoscore (IS) is a bench-to-digital pathology assay that quantifies CD3+ and cytotoxic CD8+ T-lymphocyte densities within the tumor and its invasive margin, stratifying patients into three categories: Low IS, Intermediate IS, and High IS. Multivariable Cox models stratified by center were used to assess the associations between Immunoscore and outcomes, adjusting for potential confounders, including gender, T-stage, N-stage, sidedness, and MSI. A comparison of the performance of risk prediction models was performed using the likelihood ratio test *p*-value. In uni/multivariable analyses, a High Immunoscore was significantly associated with prolonged survival of CC patients within the Asian population.

**Abstract:**

BACKGROUND: In this study, we evaluated the prognostic value of Immunoscore in patients with stage I–III colon cancer (CC) in the Asian population. These patients were originally included in an international study led by the Society for Immunotherapy of Cancer (SITC) on 2681 patients with AJCC/UICC-TNM stages I–III CC. METHODS: CD3+ and cytotoxic CD8+ T-lymphocyte densities were quantified in the tumor and invasive margin by digital pathology. The association of Immunoscore with prognosis was evaluated for time to recurrence (TTR), disease-free survival (DFS), and overall survival (OS). RESULTS: Immunoscore stratified Asian patients (n = 423) into different risk categories and was not impacted by age. Recurrence-free rates at 3 years were 78.5%, 85.2%, and 98.3% for a Low, Intermediate, and High Immunoscore, respectively (HR[Low-vs-High] = 7.26 (95% CI 1.75−30.19); *p* = 0.0064). A High Immunoscore showed a significant association with prolonged TTR, OS, and DFS (*p* < 0.05). In Cox multivariable analysis stratified by center, Immunoscore association with TTR was independent (HR[Low-vs-Int+High] = 2.22 (95% CI 1.10–4.55) *p* = 0.0269) of the patient’s gender, T-stage, N-stage, sidedness, and MSI status. A significant association of a High Immunoscore with prolonged TTR was also found among MSS (HR[Low-vs-Int+High] = 4.58 (95% CI 2.27−9.23); *p* ≤ 0.0001), stage II (HR[Low-vs-Int+High] = 2.72 (95% CI 1.35−5.51); *p* = 0.0052), low-risk stage-II (HR[Low-vs-Int+High] = 2.62 (95% CI 1.21−5.68); *p* = 0.0146), and high-risk stage II patients (HR[Low-vs-Int+High] = 3.11 (95% CI 1.39−6.91); *p* = 0.0055). CONCLUSION: A High Immunoscore is significantly associated with the prolonged survival of CC patients within the Asian population.

## 1. Introduction

The AJCC/UICC-TNM classification system based on the anatomopathological evaluation of tumors provides useful yet limited prognostic data [1]. Recent methods established to classify cancer that focus on tumor cells have demonstrated limitations in their clinical efficiency to reliably estimate outcomes [1,2]. Nevertheless, extensive studies have shed light on the adequate prognostic accuracy of the in situ immune cell infiltrate in tumors [1,3,4,5,6,7,8,9,10,11,12]. Our previous works on colorectal cancer (CRC) have shown important correlations between tumor recurrence, overall survival, and the strength of the in situ adaptive immune response [3,8,12,13,14] at the center of the tumor (CT) and its invasive margin (IM). A systematic review of 200 relevant publications depicting the role of immune cell subpopulations in the prognosis of cancer patients in 20 different cancer types showed that, in 97% of the studies, cytotoxic CD8+ T cells were associated with a good prognosis [15]. We have also reported that within specific regions of primary tumors, tumor recurrence and overall survival rates of patients with CC were mostly dependent on the presence of cytotoxic and memory T cells. In our earlier clinical study on human CRC, we showed that cytotoxic and memory T cells could predict the clinical outcome in early-stage (I/II) CRC patients. Furthermore, we revealed that the state of the local immune reaction was correlated with the histopathology-based prognostic factors of CRC. In combined tumor regions, the analysis of CD8+ cytotoxic T-lymphocyte density proved to be a better indicator of tumor recurrence than the TNM staging score [16,17,18]. This indicates that the patient’s intratumoral native adaptive immune reaction is of utmost importance for survival, strongly hinting that the immune parameters are more relevant than tumor progression and invasion classifications. This immune response was defined as the “Immunoscore” [15,19,20,21].

An international consortium of 14 care centers enrolled patients with TNM stage I–III CC and showed that Immunoscore was the first worldwide standardized consensus assay to quantify pre-existing immunity. According to these results, the consensus Immunoscore is recognized as a pertinent and powerful tool to predict the prognosis of patients [22]. The consensus Immunoscore provides a reliable assessment method for predicting the recurrence risk in CC, as confirmed by a meta-analysis of the prognostic value of Immunoscore on more than 10,000 patients [23].

Recent publications have demonstrated the prognostic value of Immunoscore in stage III CC patients and its predictive value for response to chemotherapy, thus reinforcing Immunoscore’s clinical relevance [24,25]. In the latest (5th) edition of the *WHO Digestive System Tumours classification*, the immune response evaluated with the consensus Immunoscore was defined as an “essential and desirable diagnostic criteria for colorectal cancer”. Immunoscore was also introduced into the *2020 ESMO Clinical Practice Guidelines* for CC to improve the prognosis and thus adjust the chemotherapy decision-making process in stage II and even in low-risk stage III patients. However, the clinical performance of the consensus Immunoscore in the Asian population remained to be established.

## 2. Materials and Methods

### 2.1. Patients

An international consortium composed of 14 pathology expert centers from 13 countries was initiated to evaluate the standardized Immunoscore assay in primary tumors from 2681 patients with stage I/II/III CC. The selected patients are a subset of the SITC study cohort based on an Asian population of 423 patients (Centers from Japan, China, and India). The results of this particular cohort (Asia) have not been reported before and were not shown in Pages et al. [22]. Clinical data from Asia and the complete international consortium datasets are presented in Appendix A. The outcomes of interest were time to recurrence (TTR), defined as time from surgery to disease recurrence; overall survival (OS), defined as time from surgery to death due to any cause; and disease-free survival (DFS), defined as time from surgery to disease recurrence or death from any cause. Ethical, legal, and social implications were approved by the ethical review board of each center.

### 2.2. Immunohistochemistry

At every care center, a tumor block containing the CT and IM was selected for each patient by the center’s pathologist. Two FFPE slides of 4 microns were generated per block and processed for immunohistochemistry according to a protocol recommended by the reference center and as previously described [22]. An example image of CD3 and CD8 staining is provided in Appendix A. Digital slides were obtained with a 20× magnification and a resolution of 0.45 µm/pixel.

### 2.3. Image Analysis

The stained CD3 and CD8 cell densities were determined in CT and IM regions using in-house Immunoscore software (INSERM, Paris, France). The means and distributions of staining intensities and cell densities were monitored, with an internal quality control for each slide.

### 2.4. Immunoscore Determination

For each slide, the Immunoscore was assessed: CD3 and CD8 densities in CT and IM regions were converted into percentiles, as previously described [22]. The mean of the four percentiles obtained (two markers, two regions) was calculated and translated into the Immunoscore scoring system. The Immunoscore categories were previously defined independently of clinical data [22]. These pre-defined categories were used herein, with three Immunoscore categories being defined as follows: mean percentiles of 0–25%, >25–70%, and >70–100% were Low (Lo), Intermediate (Int), and High (Hi) Immunoscore, respectively. Additionally, analyses were performed with two Immunoscore categories (Low (0–25%) and Int+Hi (25–100%) groups) and five Immunoscore categories (I0 (0–10%), I1 (>10–25%), I2 (>25–70%), I3 (>70–95%), and I4 (>95–100%) groups).

### 2.5. Monitoring of the Study

The biomarker reference center (Immunomonitoring platform, Hôpital Européen Georges Pompidou AP-HP, INSERM, Paris, France) optimized immunostaining protocols, provided the Immunoscore software user’s manual, and validated data from each cohort analyzed within each of the 14 participating centers [22]. Exclusion criteria include: missing counts at either tumor region, poor/low staining intensity (≤152 AU), damaged FFPE slides during staining, and several (>3) failed attempts at antigen retrieval. After quality control exclusion, analyses were performed on 423 Asian patients and compared to the 2681 patients included in the international consortium.

### 2.6. Statistics

Statistical analyses of demographics and disease characteristics were descriptively compared across Asia and the rest of the world and compared by t-test, Fisher’s exact test, and Chi-square test when applicable. The bivariable association between Immunoscore and time-to-event outcomes was evaluated by the log-rank test and by a participating-center-stratified Cox proportional hazards model. Multivariable Cox models stratified by center were used to assess the associations between Immunoscore and outcomes, adjusting for potential confounders (survival, R package). Model performance was assessed by Harrell’s C-statistics. The centers were used as the stratification factors, and the variables adjusted in the multivariable models were Immunoscore, gender, T-stage, N-stage, sidedness, and MSI. A comparison of the performance of risk prediction models was performed using the likelihood ratio test *p*-value. The relative importance *of* each parameter to survival risk was *assessed* using the chi-squared proportion (χ^2^) (rms, R package). An alternative measure of the survival time distribution was used, the restricted mean survival time (RMST), for two-sample comparisons (survRM2, R package) [26].

## 3. Results

### 3.1. Immune Densities and Immunoscore in Relation to the Age of the Patients

Biomarker data from 423 colon cancer patients from the Asian population (Japan, China, and India) from the AJCC/UICC-TNM stage I–III part of the consensus Immunoscore international validation study [22] were investigated. Clinical characteristics of patients from the Asian population (n = 423) were compared to the 2681 patients from the SITC international study (Appendix A). Balanced clinical characteristics were observed, with no statistical differences between cohorts in gender, T-stage, N-stage, or UICC/AJCC-TNM stages (Appendix A). However, the Asian population had slightly fewer dMMR (MSI-H) patients (6.1% vs. 11.3%), and Asian patients were more frequently younger (47.3% vs. 38.2% below 65 years old) and more frequently received chemotherapy (62.7% vs. 28%) (Appendix A). Overall, Asian patients were 54.6% male, with a mean age of 64.7 ± 12.1 years. The mean number of lymph nodes (LN) examined was 16.3 ± 9.9. Across all patients analyzed, 65 relapses (15.4% of patients) and 62 deaths (14.7% of patients) were observed. The median follow-up times (95% CI) were 73.6 (69.8–76.9), 75.0 (71.1–78.4), and 78.3 (75.0–81.6) months for TTR, DFS, and OS, respectively. The 5-year relapse or survival rates were 83.0% (79.3–86.9), 81.0% (77.2–85.0), and 87.9% (84.7–91.1) for TTR, DFS and OS, respectively (Appendix A).

Pre-defined consensus Immunoscore cut-points [22] were applied to the Asian cohort to convert CD3 and CD8 immune densities into percentiles and Immunoscore categories. The intra-tumoral densities quantified in the core of the tumor and in the invasive margin were not influenced by the age of the patients (Figure 1A). Similarly, the proportions of High-, Intermediate- and Low-Immunoscore patients were independent of the age interval (Figure 1B). Thus, Immunoscore did not significantly differ between young and elderly Asian patients.

### 3.2. Immunoscore and the Outcome of Asian Colon Cancer Patients

The prognostic value of two, three, and five categories of Immunoscore for TTR, DFS, and OS of 423 stage I–III CC patients was further evaluated in the Asian population using pre-defined cut-points (Figure 2 and Table 1).

The distribution of Immunoscore was 62.6% High+Int and 37.4% Low in two categories; 16.1% High, 46.6% Intermediate, and 37.4% Low in three categories; and 1.4%, 14.7%, 46.6%, 20.6%, and 16.8% in Immunoscore I4, I3, I2, I1, and I0, respectively. The two categories of Immunoscore enabled the identification of patients with distinct clinical outcomes for TTR (Figure 2A and Table 1). High-Immunoscore patients (63%) had a significantly longer survival for TTR (Hazard Ratio of HR_Lo/Int+Hi_ = 1.9 (1.17−3.1), *p* = 0.0097) and a higher 5-year recurrence-free rate (Hi: 86.9% (82.7–91.4%); Lo: 77% (70.5–84.1%)). The three categories of Immunoscore also enabled the identification of patients with distinct clinical outcomes for TTR (Figure 2B, Table 1). High-Immunoscore patients had a significantly longer survival for TTR (HR_Lo/Hi_ = 7.26 (1.75−30.19), *p* = 0.0064, and Trend *p* = 0.0025) and a higher 5-year recurrence-free rate (Hi: 96.3% (91.3–100%), Int: 84% (78.7–89.6%), and Lo: 77% (70.5–84.1%)). Even more striking differences were observed for the Immunoscore in five categories (Figure 2C and Table 1). High-Immunoscore patients had a significantly longer survival for TTR (HR_(I0/I3)_ = 7.75 (1.8−33.4), *p* = 0.006, and Trend *p* = 0.0019) and a higher 5-year recurrence-free rate (I4: 100% (100–100%), I3: 96% (90.7–100%), I2: 84% (78.7–89.6%), I1: 80% (71.7–89.3%), and I0: 73.5% (63.6–84.8%)) (Table 1). Forest plots are illustrated in Appendix A. Similar results were found for OS and DFS (Appendix A).

### 3.3. Immunoscore in Microsatellite-Stable (MSS) Tumors

The impact of Immunoscore was further investigated in relation to the MMR status of the patients. When stratified into two Immunoscore categories, MSS tumors were associated with an Int+Hi Immunoscore in 61.8% (152/246) of cases, while a Low Immunoscore was observed in 38.2% (94/246) of MSS tumors. For the Immunoscore in two categories, MSS patients with an Int+Hi Immunoscore had significantly longer TTR (Figure 3A and Appendix A) (HR_Lo/Int+Hi_ = 4.58 (2.27−9.23), *p* < 0.0001) and a higher 5-year recurrence-free rate (Int+Hi: 92.8% (88.7–97%) and Lo: 71.2% (62.7–81%)).

The 5-year OS survival rates were: 98.0% for Int+Hi and 80.9% for Lo; HR_Lo/Int+Hi_ = 5.3 (2.25–12.49), *p* = 0.0001. Patients with highly infiltrated MSS tumors had a survival advantage in both TTR (5-year recurrence rate, Hi: 96.9%, Int: 91.7%, and Lo: 71.2%; HR_Lo+Hi_ = 10.93 (1.49–80.49), *p* = 0.0188) and OS (5-year survival rate, Hi: 96.9%, Int: 98.3%, and Lo: 80.9%; HR_Lo+Hi_ = 7.67 (1.03–57.09), *p* = 0.0466) compared to patients with weakly infiltrated tumors. Similar results were found for DFS (Appendix A).

A similar profile was observed when three (Figure 3B) and five (Figure 3C) Immunoscore categories were applied. This analysis identified low-risk MSS patients (I4) with a significantly longer TTR compared to Immunoscore I0 MSS patients (Figure 3C). High-Immunoscore patients had a significantly longer survival for TTR (HR _I0/I3_ = 12.85 (1.68−98.28), *p =* 0.0139, and Trend *p* = 0.0005) and a higher 5-year recurrence-free rate (I4: 100% (100–100%), I3: 96.8% (90.8–100%), I2: 91.7% (86.9–96.7%), I1: 74.5% (63.9–87%), and I0: 66.6% (53.3–83.2%)). Similar results were found for OS and DFS (Appendix A).

### 3.4. Immunoscore and Time-to-Event Analysis among Patients with Stage II Colon Cancer

Stage II patients from the Asian population (n = 251) were analyzed. Low-risk patients with an Int+Hi Immunoscore presented significantly better outcomes for TTR compared to Low-Immunoscore patients (Figure 4A). The 5-year recurrence rate for patients with a High Immunoscore was 91.1% (86.4–96.1%) and only 78.3% (70.4–86.9%) for those with a Low Immunoscore. High-Immunoscore patients had a significantly longer survival for TTR (HR_Lo/Int+Hi_ = 2.72 (1.35−5.51), *p* = 0.0052). The three categories of Immunoscore also enabled the identification of patients with distinct clinical outcomes for TTR (Figure 4B). High-Immunoscore patients had a significantly longer survival for TTR (HR_Lo/Hi_ = 3.82 (0.9−16.24), *p* = 0.0697, and Trend *p* = 0.0089) and a higher 5-year recurrence-free rate (Hi: 93.5% (85.2–100%), Int: 90.5% (85–96.3%), and Lo: 78.3% (70.4–86.9%)) (Appendix A). For the Immunoscore in five categories, High-Immunoscore patients had a significantly longer survival for TTR (HR_I0/I3_ = 3.89 (0.86−17.57), Trend *p* = 0.0771) and a higher 5-year recurrence-free rate (I4: 100% (100–100%), I3: 93.2% (84.6–100%), I2: 90.5% (85–96.3%), I1: 80.4% (70.3–92.2%), and I0: 75.8% (64.3–89.4%)) (Figure 4C, Appendix A).

Among all stage II patients (n = 251), patient risk groups were defined using histopathological parameters: low risk, high risk (the extent of the primary tumor T4 or VELIPI+), and very high risk (T4 primary tumors and VELIPI+). In all risk groups (low risk (n = 224), high risk (n = 185), and very high risk (n = 27)), a High Immunoscore was associated with prolonged survival (Figure 5A–C, Appendix A).

In low-risk stage II patients, Immunoscore was also significantly associated with TTR (unadjusted HR_Lo/Int+Hi_ = 2.62 (1.21−5.68), *p* = 0.0146), and the 5-year recurrence rate was 91.6% (86.7–96.7%) for patients with a High Immunoscore and only 80.2% (72.2–89.1%) for those with a Low Immunoscore (Figure 5A). Thus, patients with a Low Immunoscore at low pathological risk were in fact at high risk of recurrence. In high-risk stage II patients (T4 or VELIPI+), Immunoscore was significantly associated with TTR (unadjusted HR_Lo/Int+Hi_ = 3.11 (1.39−6.91), *p* = 0.0055), and the 5-year recurrence rate was 91.5% (86.4–97%) for patients with a High Immunoscore and only 75.8% (66.6–86.2%) for those with a Low Immunoscore (Figure 5B). In very high-risk stage II patients, Immunoscore was also associated with TTR, with a 5-year recurrence rate of 87.5% (72.7–100%) for patients with a High Immunoscore and only 63.6% (40.7–99.5%) for those with a Low Immunoscore (Figure 5C). Strikingly, patients with high-risk or very high-risk stage II and a High Immunoscore had a good outcome, similar to the rest of the Stage II cohort with lower risk (Figure 5D,E; Appendix A).

### 3.5. Performance of Immunoscore in Multivariable Analyses

Cox multivariable analyses adjusted for Immunoscore, age, gender, T-stage, N-stage, sidedness, and MSI and stratified by city center revealed the significant prognostic value of Immunoscore (Figure 6). In a multivariable model, age, gender, T-stage (T3 vs. T1–2), N-stage (N1 vs. N0), sidedness, and MSI were not significant. Among tumor-related parameters, only T-stage ((T4 vs. T1–2), HR = 14.35 (1.73–119.27), *p* = 0.0137) and N-stage ((N2 vs. N0), HR = 2.35 (1.13–4.89), *p* = 0.0223) were significant for TTR (Appendix A). The Immunoscore in two categories (Lo/Int+Hi) was significant in Cox multivariable analyses, with HR = 2.22 (1.10–4.55), *p* = 0.0269. Cox multivariable analyses for OS and DFS showed similar results, with significant *p*-values for Immunoscore in OS (*p* = 0.0304) and DFS (*p* = 0.0516).

The power of Immunoscore in OS was also evaluated using the contribution to risk. Indeed, variables with the most important relative contribution to the risk of death (Chi^2^ proportion) were: Immunoscore (25%), T-Stage (28%), age (18%), MSI (17%), N-Stage (12%), and gender (1%). In multivariable analysis, the only variables shown to be of significant predictive value were Immunoscore, T-stage, and N-stage. Moreover, the predictive power of Immunoscore for recurrence (likelihood ratio test, *p* < 0.0001) and death (likelihood ratio test, *p* < 0.0001) was further strengthened when adding it to a model combining all clinical variables.

## 4. Discussion

The major prognostic impact of the immune contexture has been demonstrated in several studies [27,28,29]. The powerful assessment of immune cells in the tumor using digital pathology led to the international validation of the Immunoscore assay in stage I/II/III CC [22], as well as in stage III patients [24,25,30], and in two randomized phase 3 clinical trials [24,25]. The prognostic impact of the tumor microenvironment and Immunoscore has been clearly established, from pre-cancer lesions [31] to primary tumors [3,7,8,12,14,22,27,32] to metastasis [29,33,34,35,36]. The study complied with the STARD reporting guidelines (Appendix A). Beyond the results obtained for stages I/II/III [3,8,22], for localized cancers [14,22], and for metastatic diseases (stage IV) [29,33,34,35,36,37], the relevance of the consensus Immunoscore in the Asian population remained to be established. Based on immune parameters alone, we highlighted the ability of the consensus Immunoscore to accurately layer all patients and, on an anatomopathological basis, defined high- and low-risk patients with significant differences in clinical survival. Interestingly, one of the most used tools in clinical oncology (i.e., MSI status) was shown to be dependent on the Immunoscore, as presented in our Cox multivariable analyses. We also found that the local intra-tumoral immune environment was not affected by patient age in the Asian population, thus contrasting with previous reports suggesting that peripheral and intra-tumoral immunity were known to decline over time [30,38].

In addition to its strong prognostic value, Immunoscore also predicted the response to chemotherapy in an international cohort study [30] and in a randomized phase 3 clinical trial [24]. Many guidelines include chemotherapy as a potent treatment for all stage III CC. Indeed, following surgical resection, the risk of death decreases by 10% to 15% when patients are treated with 5-FU and by 20% when treated with the oxaliplatin–fluoropyrimidine combination [39,40,41]. However, in stage III CC, a mere 20% of patients can benefit from adjuvant chemotherapy (AC), leaving 80% of patients susceptible to unneeded toxicity. In fact, 50% of those patients could be cured by surgery alone, and even with AC treatment, 30% of patients experience events of recurrence that lead to death within 2-3 years [42].

Previously, it has been shown that chemotherapy’s anti-tumoral activity is tightly linked to the immune response within the tumor, as it can modulate the immune system both positively and negatively [43,44,45,46]. Accordingly, Immunoscore was developed to help segregate patients who could benefit from chemotherapy. In this study, we showed that patients with better pre-built immunity (i.e., Intermediate and High Immunoscores) do benefit the most from chemotherapy, whereas Low-Immunoscore patients fall short in response to chemotherapy. Similar findings were observed in all stage III and low-risk and high-risk stage III patients, suggesting that effective chemotherapy partly relies on the modulation of the immune system and the high density of pre-existing tumor-infiltrating T cells, a hallmark of immune surveillance. Interestingly, none of the few patients with the highest Immunoscores (I4) relapsed, even when they were not treated with chemotherapy [30], supporting the idea of sparing these patients from unnecessary chemotherapy.

A limitation of the study might be the heterogeneity of the patient population, having come from three large countries, namely, China, India, and Japan. However, this non-randomized approach aimed at enhancing the robustness of the consensus Immunoscore within the Asian population. In particular, the use of chemotherapy and its impact on survival cannot be analyzed in an overall population including stages I, II, and III, since these patients have different outcomes and do not receive chemotherapy to equal extents (stage I does not receive chemotherapy, stage II may be provided with chemotherapy depending on risk factors, and stage III should undergo chemotherapy based on international recommendations).

For this aim, subgroup analysis has to be performed. However, in our present study, the sample size did not allow us to appropriately evaluate the benefit of chemotherapy.

Within stage III (n = 105 patients), only 11 did not receive chemotherapy. This is related to different reasons, such as patient refusal or a critical health condition. Sub-dividing these 11 patients into Immunoscore categories would not lead to ultimate statistical conclusions. Indeed, there was no significant difference in survival between patients receiving or not receiving chemotherapy in this cohort.

Within stage II (n = 251 patients), 148 received chemotherapy. Since the use of chemotherapy is not recommended for all stage II patients, decisions were made based on risk factors. So far, no randomized studies have shown a significant benefit of chemotherapy within the subgroup of stage II patients. Thus, the limited number of patients analyzed herein would not provide statistical conclusions, and much larger stage II groups should be analyzed to reach significant conclusions.

Moving forward, it will be important to further validate the standardized Immunoscore assay in randomized clinical trials of stage II and/or III CC treated with adjuvant chemotherapy in the Asian population [47,48].

A meta-analysis of the prognostic value of Immunoscore conducted on more than 10,000 patients confirmed that Immunoscore provided a reliable estimate of the recurrence risk in colon cancer [23].

## 5. Conclusions

The present study further enhances the clinical utility of Immunoscore in Asian CC patients. Developed as an in vitro diagnostic test, the Immunoscore assay is available in FDA CLIA-certified laboratories and in China for clinical use (CE-IVD). Moreover, the 5th edition of the *WHO Digestive System Tumours* classification introduced, for the first time, the immune response as “essential and desirable diagnostic criteria for colorectal cancer” while citing the consensus Immunoscore as the “best clinical evidence in colon cancer”. In fact, Immunoscore was also introduced in the 2020 *ESMO Clinical Practice Guidelines* for colon cancer patient support, allowing physicians to refine the prognosis and thus adjust the chemotherapy decision-making process in stage II and even in low-risk stage III patients [49]. Recently, Immunoscore was introduced into the Pan-Asian-adapted *ESMO Clinical Practice Guidelines* for the diagnosis, treatment, and follow-up of patients with localized colon cancer. Immunoscore was considered for its full-range indication in colon cancers, stage II and stage III, with the inclusion of all risk groups [50]. Supported by the multicentric international SITC study, the results of Immunoscore in the Asian population and its recent inclusion in the above-cited guidelines argue for the benefit of implementing Immunoscore in routine clinical practice as well as its introduction in other international guidelines. This would allow patients and physicians to benefit from this powerful predictive tool in colorectal cancer support.

## 6. Patents

J.G., F.P. and B.M. have patents associated with the immune prognostic biomarkers. Immunoscore^®^ is a registered trademark owned by the National Institute of Health and Medical Research (INSERM) and licensed to Veracyte. Michael Roehrl is a member of the Scientific Advisory Boards of Azenta and Universal DX. All other authors declare no conflict of interest.

## Figures and Tables

**Figure 1 cancers-14-04346-f001:**
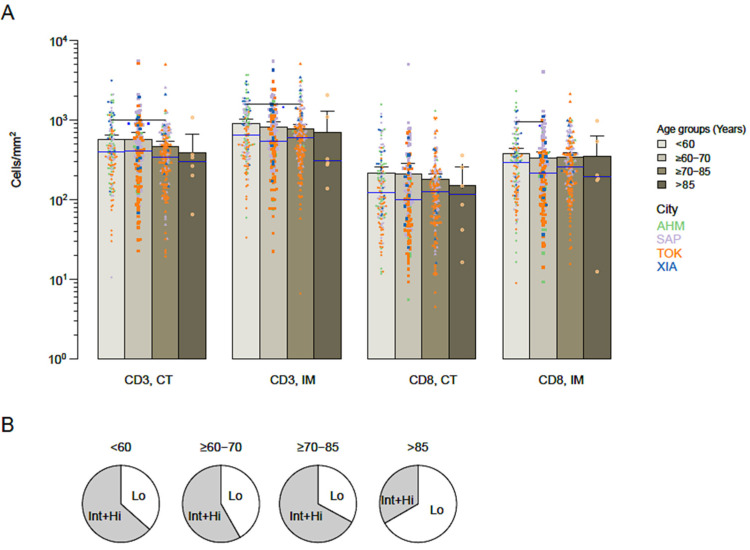
The immune infiltrate and Immunoscore in patients based on age. (**A**) Patients were grouped according to their age: <60 years (orange), ≥60–70 (yellow), ≥70–85 (purple), and ≥85 (blue). Immune densities of CD3 and CD8 quantified in the tumor core (CT) and invasive margin (IM). Each dot represents the mean whole-slide quantification for one patient. (**B**) Distribution of Intermediate and High (Int+Hi) versus Low (Lo) Immunoscore in age-based patient groups. AHM (Ahmedabad, India); SAP (Sapporo, Japan); TOK (Tokyo, Japan); XIA (Xi’an, China).

**Figure 2 cancers-14-04346-f002:**
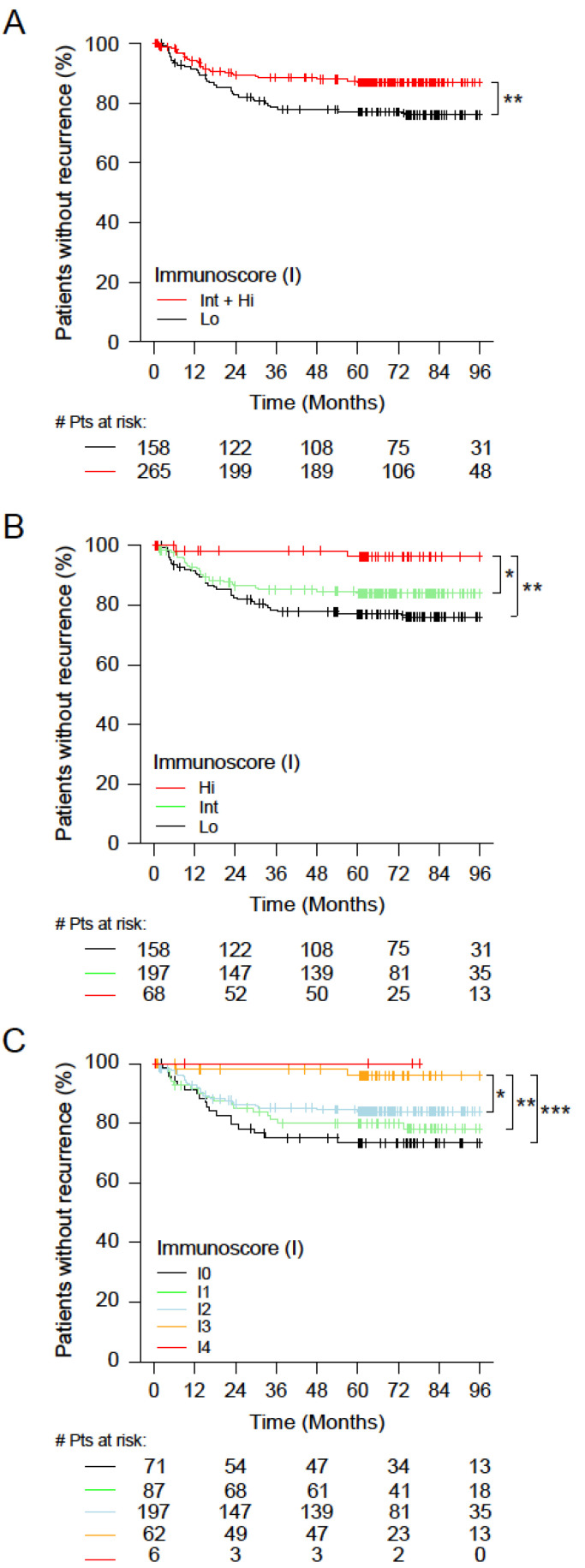
The outcomes of stage I–III colon cancer patients according to Immunoscore. (**A**–**C**) Kaplan–Meier curves of Immunoscore for stage I–III patients are shown for time to recurrence (TTR). (**A**) Two Immunoscore categories: Lo (0–25%, black) and Int+Hi (>25–100%, red). (**B**) Three Immunoscore categories: Lo (0–25%, black), Int (>25–70%, green), and Hi (>70–100%, red). (**C**) Five Immunoscore categories: I0 (0–10%, black), I1 (>10–25%, green), I2 (>25–70%, azure), I3 (>70–95%, orange), and I4 (>95–100%, red). Significant log-rank *p*-values are marked as *** *p* < 0.005, ** 0.005 ≥ *p* < 0.01, and * 0.01 ≥ *p* < 0.05.

**Figure 3 cancers-14-04346-f003:**
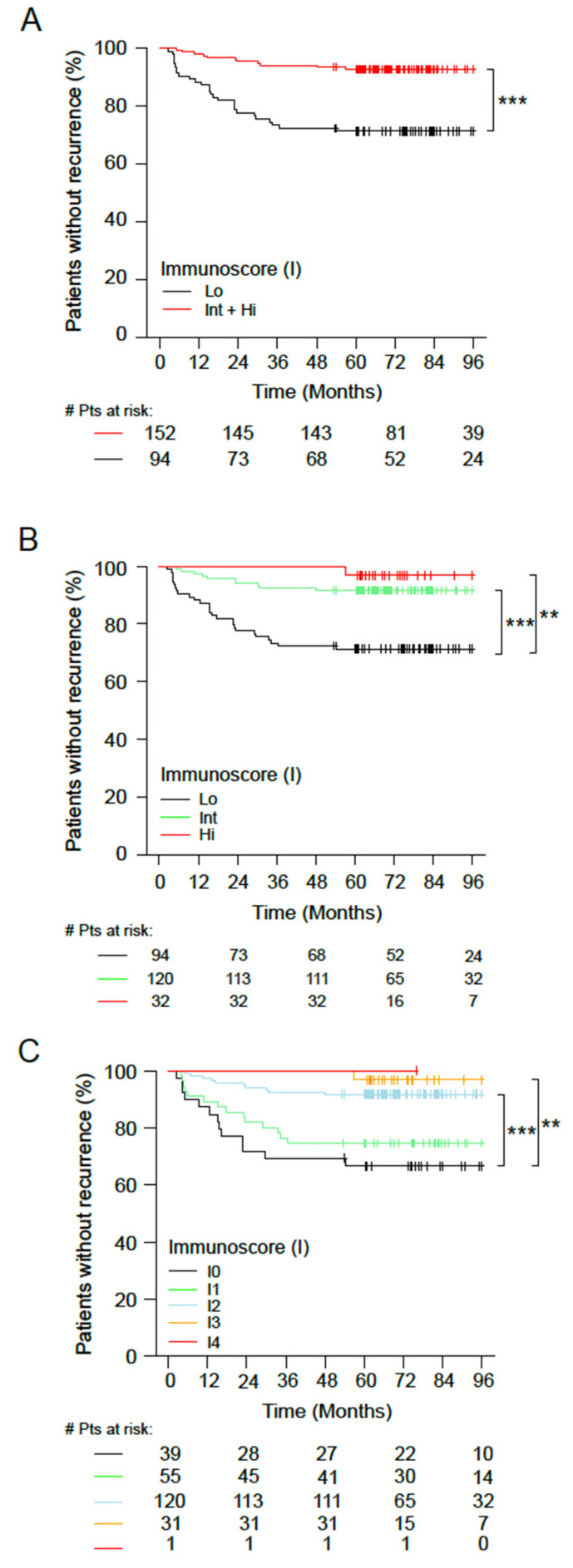
The outcomes of stage I–III microsatellite-stable (MSS) colon cancer according to Immunoscore. (**A**–**C**) Kaplan–Meier curves of Immunoscore for stage I–III MSS patients are shown for TTR. (**A**) Two Immunoscore categories: Lo (0–25%, black) and Int+Hi (>25–100%, red). **(B**) Three Immunoscore categories: Lo (0–25%, black), Int (>25–70 %, green), and Hi (>70–100 %, red). (**C**) Five Immunoscore categories: I0 (0–10%, black), I1 (>10–25%, green), I2 (>25–70%, azure), I3 (>70–95%, orange), and I4 (>95–100%, red). Significant log-rank *p*-values are marked as *** *p* < 0.005 and ** 0.005 ≥ *p* < 0.01.

**Figure 4 cancers-14-04346-f004:**
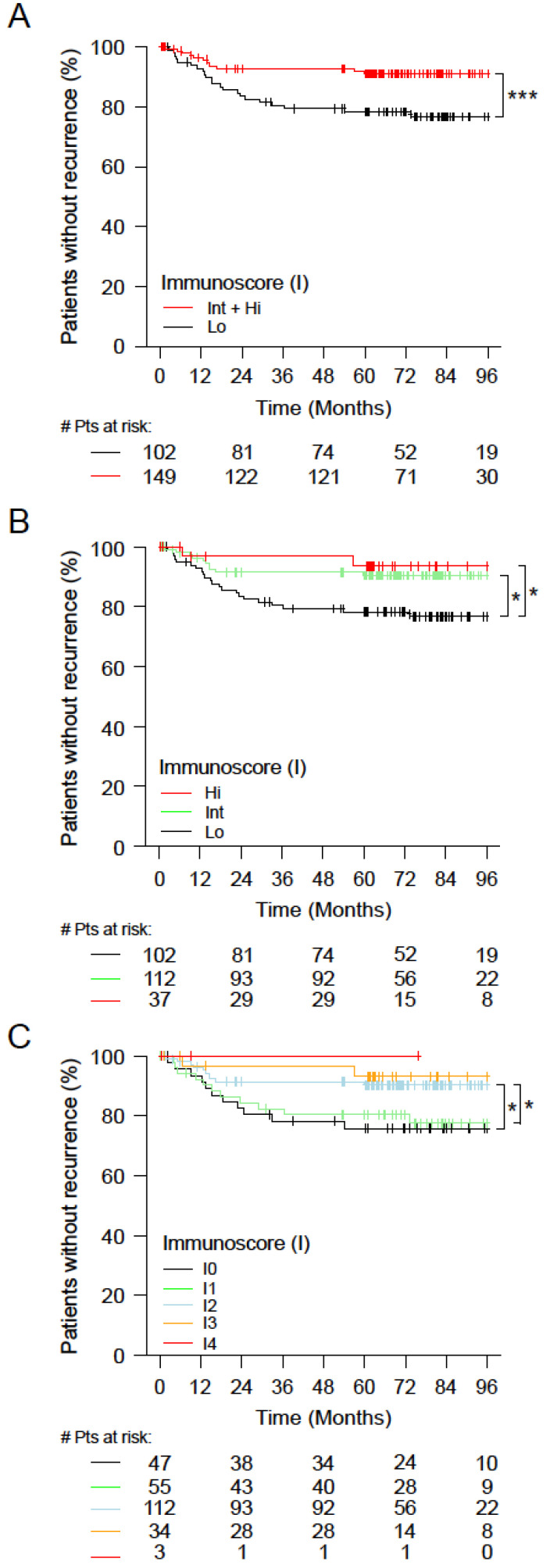
The outcomes of stage II colon cancer patients according to Immunoscore. (**A**–**C**) Kaplan–Meier curves of Immunoscore for stage II patients are shown for time to recurrence (TTR). (**A**) Two Immunoscore categories: Lo (0–25%, black) and Int+Hi (>25–100%, red). (**B**) Three Immunoscore categories: Lo (0–25%, black), Int (>25–70%, green), and Hi (>70–100%, red). (**C**) Five Immunoscore categories: I0 (0–10%, black), I1 (>10–25%, green), I2 (>25–70%, azure), I3 (>70–95%, orange), and I4 (>95–100%, red). Significant log-rank *p*-values are marked as *** *p* < 0.005 and * 0.01 ≥ *p* < 0.05.

**Figure 5 cancers-14-04346-f005:**
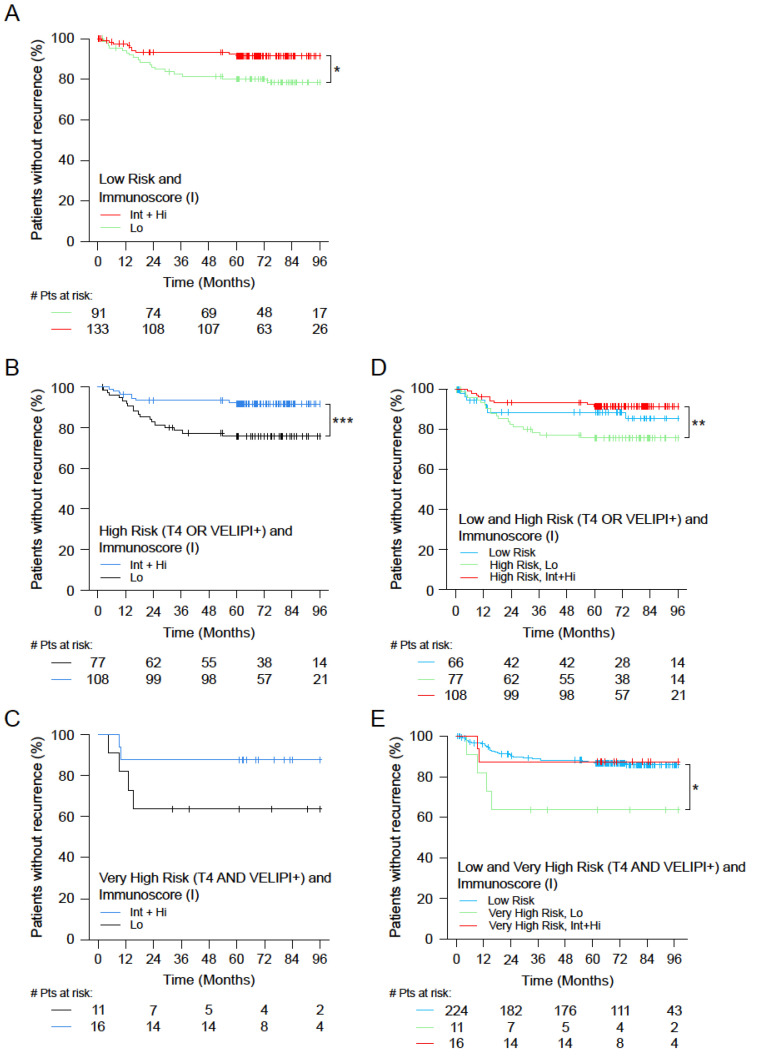
The impact of Immunoscore on the outcome of low- and high-risk colon cancer patients. Kaplan–Meier curves of Low (Lo, 0–25%) and Intermediate+High (Int+Hi, >25–100%) Immunoscore are shown for TTR (**A**–**C**). (**A**) Low-risk patients, Immunoscore Lo (green) and Int+Hi (red). (**B**) High-risk patients (T4 and VELIPI+), Immunoscore Lo (black) and Int+Hi (blue). (**C**) Very high-risk patients (T4 or VELIPI+), Immunoscore Lo (black) and Int+Hi (blue). (**D**) Low and high-risk patients (T4 or VELIPI+). (**E**) Low and very high-risk patients (T4 and VELIPI+). Significant log-rank *p*-values are marked as *** *p* < 0.005, ** 0.005 ≥ *p* < 0.01, and * 0.01 ≥ *p* < 0.05.

**Figure 6 cancers-14-04346-f006:**
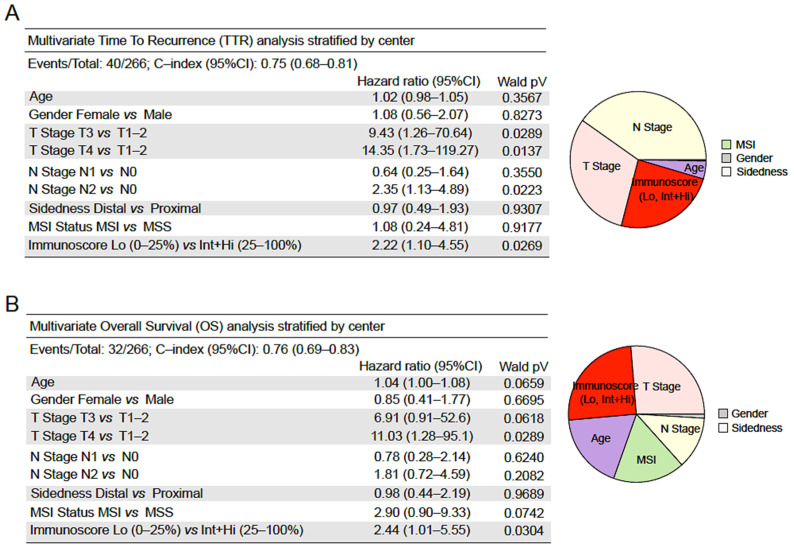
Clinical performance of tumor- and immune-related risk parameters. Cox multivariable regression analysis of TTR (**A**) and OS (**B**) combining clinical parameters and Immunoscore with two (0–25% and 25–100%) categories. Clinical parameters: age, gender, T-stage, N-stage, sidedness, and MSI status. The relative importance of each risk parameter to survival risk using the χ2 proportion test for clinical parameters and Immunoscore (right).

**Table 1 cancers-14-04346-t001:** Stage I–III bivariable analysis for clinical parameters for TTR.

		Time to Recurrence (TTR)						
	Number of	Rate at		Unadjusted			RMST	
	Patients (%)	3 yr % (95% CI)	5 yr % (95% CI)	HR (95% CI)	*p*-Value *	C-Index (95% CI)	Rel. Months (95% CI)	*p*-Value **
**Age at surgery (5 groups)**						0.55 (0.49−0.62)		
<60	134 (31.7)	82.4 (75.7–89.8)	81.3 (74.3–88.9)	1.0 (reference)			0.0 (reference)	
≥60–70	134 (31.7)	85.3 (79.2–91.8)	83.5 (77.2–90.4)	0.87 (0.47−1.59)	0.6500		2.2 (−5.1–9.6)	0.5517
≥70–85	149 (35.2)	86.4 (80.8–92.2)	85.6 (80–91.7)	0.72 (0.39−1.32)	0.2854		3.9 (−3.2–10.9)	0.2832
>85	6 (1.4)	66.7 (37.9–100)	50 (22.5–100)	2.85 (0.85−9.56)	0.0897		−15.8 (−43.3–11.7)	0.2609
**Gender**						0.5 (0.44−0.56)		
Male	231 (54.6)	84.2 (79.3–89.4)	83.6 (78.6–88.9)	1.0 (reference)			0.0 (reference)	
Female	192 (45.4)	84.9 (79.7–90.3)	82.4 (77–88.3)	1.03 (0.63−1.67)	0.9133		−0.4 (−13.1–12.3)	0.9520
**T stage**						0.62 (0.56−0.67)		
T1	20 (4.7)	100 (100–100)	100 (100–100)	1.0 (reference)			0.0 (reference)	
T2	53 (12.5)	93.7 (87–100)	93.7 (87–100)	Inf (0-Inf)	NA		−8.2 (−17.1–0.8)	0.0737
T3	289 (68.3)	84.9 (80.7–89.4)	82.9 (78.4–87.6)	Inf (0-Inf)	NA		−20.5 (−26–15)	<0.0001
T4	61 (14.4)	68.9 (57.6–82.4)	68.9 (57.6–82.4)	Inf (0-Inf)	NA		−38.7 (−54.1–23.4)	<0.0001
**N stage**						0.64 (0.58−0.71)		
N0	318 (75.2)	89.8 (86.4–93.3)	88.3 (84.7–92.1)	1.0 (reference)			0.0 (reference)	
N1	66 (15.6)	79.2 (69.3–90.4)	79.2 (69.3–90.4)	1.94 (1.01−3.74)	0.0477		−11.3 (−24.6–1.9)	0.0944
N2	39 (9.2)	38.5 (23.5–62.8)	34.2 (19.9–58.8)	6.91 (3.89−12.27)	<.0001		−59.5 (−80.5–38.6)	<0.0001
**AJCC/UICC-TNM stage**						0.67 (0.62−0.72)		
I	67 (15.8)	98.3 (95.1–100)	98.3 (95.1–100)	1.0 (reference)			0.0 (reference)	
II	251 (59.3)	87.6 (83.4–91.9)	85.7 (81.3–90.4)	9.34 (1.28−68.23)	0.0277		−16.9 (−24.4–9.5)	<0.0001
III	105 (24.8)	66.2 (56.8–77.1)	64.9 (55.4–76)	25.62 (3.49−187.95)	0.0014		−44.3 (−58.5–30.1)	<0.0001
**Differentiation grade**						0.62 (0.56−0.67)		
Well	118 (28.2)	91.3 (86.4–96.6)	91.3 (86.4–96.6)	1.0 (reference)			0.0 (reference)	
Moderate	261 (62.4)	83.3 (78.6–88.3)	81.4 (76.5–86.7)	2.35 (1.18−4.67)	0.0152		−14.9 (−25.1–4.7)	0.0043
Poor–undiff.	39 (9.3)	68.4 (53.9–86.8)	64.1 (49–84)	5.15 (2.19−12.14)	0.0002		−39.3 (−64.9–13.6)	0.0027
**Proximal vs. Distal Primary (Tumor)**						0.51 (0.45−0.57)		
Proximal	184 (44.2)	84.2 (78.7–90.1)	83.5 (78–89.5)	1.0 (reference)			0.0 (reference)	
Distal	232 (55.8)	84.7 (80–89.6)	82.6 (77.7–87.9)	1.07 (0.65−1.77)	0.7848		−1.6 (−14.5–11.3)	0.8042
VELIPI						0.53 (0.48−0.58)		
NO	122 (28.8)	88.3 (82.1–95.1)	88.3 (82.1–95.1)	1.0 (reference)			0.0 (reference)	
YES	301 (71.2)	83.3 (79–87.7)	81.4 (77–86.1)	1.44 (0.77−2.7)	0.2514		−9.1 (−24.4–6.2)	0.2433
**Mucinous colloid type**						0.51 (0.48−0.54)		
NO	367 (95.3)	84.9 (81.2–88.7)	83.3 (79.5–87.3)	1.0 (reference)			0.0 (reference)	
YES	18 (4.7)	75.1 (56.6–99.7)	75.1 (56.6–99.7)	1.54 (0.56−4.23)	0.4061		−11.9 (−43.7–19.9)	0.4632
**MSI Status (Derived)**						0.52 (0.49−0.56)		
MSS	246 (90.4)	86.2 (82–90.6)	84.5 (80.1–89.2)	1.0 (reference)			0.0 (reference)	
MSI–H	26 (9.6)	92.3 (82.6–100)	92.3 (82.6–100)	0.48 (0.12−2.01)	0.3168		9.6 (−5.7–24.8)	0.2178
**Adjuvant chemotherapy**						0.57 (0.5-0.63)		
NO	146 (35.5)	89.9 (85–95.1)	89.9 (85–95.1)	1.0 (reference)			0.0 (reference)	
YES	265 (64.5)	81.8 (76.9–86.9)	79.4 (74.2–84.8)	2 (1.12−3.57)	0.0198		−15.4 (−27.8–3.1)	0.0145
**Immunoscore Lo vs. Int+Hi**						0.58 (0.52−0.64)		
Lo (0–25%)	158 (37.4)	78.5 (72.1–85.4)	77 (70.5–84.1)	1.9 (1.17−3.1)	0.0097		−19.3 (−34.1–4.5)	0.0106
Int+Hi (25–100%)	265 (62.6)	88.4 (84.3–92.6)	86.9 (82.7–91.4)	1.0 (reference)			0.0 (reference)	
**Immunoscore Lo vs. Int vs. Hi**						0.6 (0.55−0.66)		
Lo (0–25%)	158 (37.4)	78.5 (72.1–85.4)	77 (70.5–84.1)	7.26 (1.75−30.19)	0.0064		−33.5 (−47.2–19.9)	<0.0001
Int (25–70%)	197 (46.6)	85.2 (80.1–90.6)	84 (78.7–89.6)	4.77 (1.14−20.04)	0.0327		−21.1 (−32.9–9.3)	0.0005
Hi (70–100%)	68 (16.1)	98.3 (95–100)	96.3 (91.3–100)	1.0 (reference)			0.0 (reference)	
**Immunoscore**						0.61 (0.55−0.67)		
I0 (0–10%)	71 (16.8)	75.1 (65.5–86.1)	73.5 (63.6–84.8)	7.75 (1.8−33.4)	0.0060		−16.1 (−22.7–9.6)	<0.0001
I1 (10–25%)	87 (20.6)	81.3 (73.2–90.3)	80 (71.7–89.3)	6.05 (1.4−26.18)	0.0161		−12.4 (−17.9–6.9)	<0.0001
I2 (25–70%)	197 (46.6)	85.2 (80.1–90.6)	84 (78.7–89.6)	4.48 (1.07−18.82)	0.0404		−10 (−13.4–6.5)	<0.0001
I3 (70–95%)	62 (14.7)	98.1 (94.6–100)	96 (90.7–100)	1.0 (reference)			−1.8 (−4.5–0.9)	0.1969
I4 (95–100%)	6 (1.4)	100 (100–100)	100 (100–100)	Inf (0-Inf)	NA		0.0 (reference)	
**Immunoscore Lo vs. Int+Hi and High risk (T4 and VELIPI+) vs. Low risk (all others)**						0.58 (0.52−0.65)		
0–25% High Risk	11 (2.6)	63.6 (40.7–99.5)	63.6 (40.7–99.5)	3.48 (1.22−9.91)	0.0198		−27.1 (−59.4–5.3)	0.1007
0–25% Low Risk	147 (34.8)	79.7 (73.2–86.7)	78.1 (71.5–85.4)	1.79 (1.08−2.99)	0.0250		−9.7 (−18.5–0.9)	0.0302
25–100% High Risk	16 (3.8)	87.5 (72.7–100)	87.5 (72.7–100)	0.97 (0.23−4.06)	0.9643		−0.4 (−19.4–18.6)	0.9678
25–100% Low Risk	249 (58.9)	88.4 (84.2–92.8)	86.9 (82.4–91.5)	1.0 (reference)			0.0 (reference)	

* Wald *p*-value. ** Restricted mean survival time (RMST) *p*-value. MSS: proficient mismatch repair (pMMR).

## Data Availability

Data are available upon request to the corresponding author.

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
