# Peer review of "Clinical Performance of the Consensus Immunoscore in Colon Cancer in the Asian Population from the Multicenter International SITC Study"

_cancers, 2022, doi:10.3390/cancers14184346_

Round 1

Reviewer 1 Report

The authors provide a very nice study about Immunoscore in a subset of Asian patients with colon cancer. Two main remarks:

- Is there a reason the 3-category panel was not checked for MSS cancers?

- It is important to highlight the use of adjuvant chemotherapy and its association with survival. 

- If possible, mention any correlation with circulating tumor dna assays if the patients had this done too. 

Author Response

Dear reviewer,

Thank you for the careful review of our manuscript that we are submitting to Cancers. We haven taken your comments into consideration and made the necessary modifications to the manuscript. Please find in attachment our answers to your questions.

Best regards.

Jerome Galon. 

Reviewer 2 Report

Mlecnik, B et al.: Clinical performance of the consensus Immunoscore in colon cancer in the Asian population from the multicenter international SITC study
Journal: Cancers
. (MS No. Cancers-1861185)

The authors demonstrated that a high-Immunocore is significantly associated with prolonged survival of colon cancer patients within the Asian population like that of the European population. This analysis will be of great clinical benefit because it will allow us to select a subset of Stage 3 colorectal cancer patients who will require adjuvant chemotherapy. The paper was also analyzed and concluded for a cohort of patients from an independent hospital, so the results obtained are very robust. This journal has been concluded with excellent analyses of independent institutes.However, there are some points left that require clarification before the manuscript can be considered suitable for publication.

Minor issues:

1.        The authors only analyzed MSS tumors. How about Immunocore in MSI-H tumors? Even if there is no difference, the authors should show the results.

2.        The original photos of the Immunocore on digital slides should be presented because this information will provide evidence to make a clinical decision.

Author Response

Dear reviewer,

We would like to thank you for your careful review of our manuscript that we are submitting to Cancers. We have taken into consideration your comments and suggestions and made the necessary modifications to the manuscript.

Please find in attachment our answers to your questions.

Best regards.

Jerome Galon
